# It’s Time for Entropic Clocks: The Roles of Random Chain Protein Sequences in Timing Ion Channel Processes Underlying Action Potential Properties

**DOI:** 10.3390/e25091351

**Published:** 2023-09-17

**Authors:** Esraa Nsasra, Irit Dahan, Jerry Eichler, Ofer Yifrach

**Affiliations:** Department of Life Sciences, School of Brain Sciences and Cognition, Ben-Gurion University of the Negev, P.O. Box 653, Beer Sheva 84105, Israel; esraan@post.bgu.ac.il (E.N.); jeichler@bgu.ac.il (J.E.)

**Keywords:** action potential, alternative splicing, ball and chain, channel clustering, density regulation, entropic chains, hetero-oligomerization, intrinsic disorder, potassium channels, scaffold proteins

## Abstract

In recent years, it has become clear that intrinsically disordered protein segments play diverse functional roles in many cellular processes, thus leading to a reassessment of the classical structure–function paradigm. One class of intrinsically disordered protein segments is entropic clocks, corresponding to unstructured random protein chains involved in timing cellular processes. Such clocks were shown to modulate ion channel processes underlying action potential generation, propagation, and transmission. In this review, we survey the role of entropic clocks in timing intra- and inter-molecular binding events of voltage-activated potassium channels involved in gating and clustering processes, respectively, and where both are known to occur according to a similar ‘ball and chain’ mechanism. We begin by delineating the thermodynamic and timing signatures of a ‘ball and chain’-based binding mechanism involving entropic clocks, followed by a detailed analysis of the use of such a mechanism in the prototypical *Shaker* voltage-activated K^+^ channel model protein, with particular emphasis on ion channel clustering. We demonstrate how ‘chain’-level alternative splicing of the Kv channel gene modulates entropic clock-based ‘ball and chain’ inactivation and clustering channel functions. As such, the Kv channel model system exemplifies how linkage between alternative splicing and intrinsic disorder enables the functional diversity underlying changes in electrical signaling.

## 1. Introduction

Electrical signaling in the nervous system depends on action potential generation, propagation, and transmission [1]. These processes rely on precisely timed events that depend on gating transitions of voltage-dependent Na^+^ and K^+^ channels [1] clustered in multiple copies at unique membrane sites, such as the initial segment of an axon, nodes of Ranvier, or the post-synaptic density (PSD) [2,3]. Changes in either ionic current shape or density, indicative of changes in temporal and spatial dimensions, respectively, affect action potential shape and frequency [4,5,6,7,8,9] and can result in altered neuron firing [8]. Defining the molecular mechanisms that underlie channel gating and clustering is thus central for cellular-level understanding of electrical signaling, as well as its physiological importance [10].

Recently, it has become clear of late that intrinsically disordered protein segments, i.e., extended protein segments that lack a stable three-dimensional structure and behave as random chains [11], play important roles in electrical signaling. In particular, it has been shown that distinct ion channel proteins employ such extended protein segments to regulate channel gating [12] and clustering [13], at both the molecular and cellular levels [10]. A well-studied example of the roles of extended random chains in electrical signaling refers to the prototypical *Shaker* voltage-activated potassium channel (Kv), known to be involved in shaping action potentials and transmitting electrical signals across the neuromuscular junction. The *Shaker* Kv channel employs two distinct intrinsically disordered protein segments located at either the N or C terminus and which function as entropic clock chains [10]. These cytoplasmic N- and C-unstructured tails are involved in fast inactivation gating and in channel clustering, respectively, as described by a ‘ball and chain’ binding mechanism (Figure 1).

The ‘ball and chain’ mechanism was originally introduced by Armstrong and Bezanilla to describe how fast inactivation regulates Na^+^ currents through voltage-activated sodium channels [16]. Later, Aldrich and colleagues have shown that the same mechanism applies to describe K^+^ current blockage through the *Shaker* Kv channel [17,18]. Inactivation refers to the process by which sodium/potassium currents passing through the appropriate channel are blocked or decreased over time, even when the depolarizing voltage stimulus that led to channel opening remains [19,20]. With respect to the *Shaker* Kv channel, Aldrich and colleagues noticed that amino acid deletions at the cytoplasmic N-terminal tail of the channel dramatically affected inactivation gating. Furthermore, these deletions caused accelerated entry of the channels into the inactivated state [17,18] but not the rate of recovery of the channel from inactivation [21]. Based on these and other observations, they formulated the ‘ball and chain’ mechanism, according to which the N-terminal tail of the *Shaker* Kv channel comprises a 20 amino acid-sequence motif involved in channel blocking, termed the ‘ball’, linked to a long unstructured ‘chain’ (Figure 1a). The extended ‘chain’ provides the degrees of freedom required for the ‘ball’ to reach and bind to its receptor site at the inner cavity at the open pore domain of the channel, thereby timing potassium current blockage [12,18,21,22,23,24,25]. This blockage reaction is intra-molecular, as the ‘ball and chain’ sequence is attached to the channel that it blocks [12]. More than 10 years later, Dunker and colleagues, who were the first to systematically explore the various functional roles played by intrinsically disordered protein segments [11], described how the ‘ball and chain’ extended sequence of the Kv channel acts as an entropic clock. This clock serves to time the binding reaction, with the length of the ‘chain’ corresponding to the clock hands. Our lab has recently shown how the *Shaker* Kv channel protein exploits an additional entropic clock chain for binding to membrane-associated PSD-95 synaptic scaffold proteins [13] (Figure 1b). This interaction is responsible for the clustering of many Kv channel molecules at unique membrane sites [26,27,28,29], a crucial event for efficient electrical signaling [29,30,31]. This interaction, however, involves the cytoplasmically oriented C-terminal tail of the channel and is, in essence, inter-molecular [13]. Accordingly, we next describe these entropic clocks and their physiological relevance in electrical signaling.

## 2. Thermodynamic and Timing Signatures of Entropic Clock-Based ‘Ball and Chain’ Binding Mechanism

What is the signature of a ‘ball and chain’ mechanism? Studies on the N-terminal-based inactivation and the C-terminal-based clustering processes performed by the *Shaker* Kv channel argue that, for a ‘ball and chain’ mechanism to apply, three criteria must be met. First, the putative ‘ball and chain’ entropic chain sequence must be shown to indeed be an intrinsically disordered random chain. Second, the binding of the ‘ball’ sequence to its receptor site should be entropy-controlled, with the ‘chain’ length determining the binding energy of the reaction (see below, Figure 2, and reference [10]). The third and last criterion refers to the timing aspect of the binding reaction. One needs to show that only the association rate constant (*k_a_*) is affected by ‘chain’ length, which has no effect on the dissociation rate constant (*k_d_*). This requirement is also reflected in the invariance of the *k_a_* vs. *k_d_* plot and is the ‘timing’ criterion for entropic clock chains involved in binding events [10].

With respect to the first criterion that argues for a ‘ball and chain’ mechanism, both the N- and C-termini of the *Shaker* Kv channel were shown to be intrinsically disordered, using a reductionist approach involving biochemical and biophysical analyses [10]. Second, we and others have shown that the binding reactions of both the N- and C-terminal *Shaker* channel chains are entropy-controlled, as dictated by the appropriate terminal chain length [12,13]. Specifically, the shorter the chain, the higher the observed affinity for the partner protein. Thus, both the N- and C-terminal tails are entropic chains, the lengths of which determine the affinity of the ‘ball’ to its receptor site. This is portrayed in Figure 2a for the case of scaffold protein binding to the C-terminal ‘chain’ of the *Shaker* channel. Furthermore, for a pure ‘ball and chain’ binding reaction, the ‘ball’ is expected to contribute only to the enthalpy of the binding reaction (ΔH) (upon interacting with its receptor site), whereas the ‘chain’ contributes to the entropy (ΔS) of the reaction. Thus, invariance between the entropy and enthalpy of the binding reaction is expected for perturbations in the ‘ball and chain’ binding sequence, in contrast to many other binding studies reporting enthalpy–entropy compensation upon perturbation of either the protein or its ligand [10,32]. The ΔH-ΔS invariance plot is thus a thermodynamic signature of an entropic clock-based ‘ball and chain’ mechanism. This indeed was found to be the case, as the entropy of the binding reaction, but not the enthalpy, was shown to be dependent on C-terminal ‘chain’ length [13] (Figure 2b). Similar results have yet to be reported for the case of the N-terminal inactivation channel ‘chain’. The third ‘timing’ criterion was also met for both the N- and C-terminal channel chains. An entropic clock function requires that ‘chain’ length primarily affects the kinetics of the binding phase and not the kinetics of unbinding (dissociation) [11,13,33]. As an example, see Figure 2c demonstrating the dependence of the association rate constant between the Kv channel and the PSD-95 scaffold protein on C-terminal ‘chain’ length. Such dependence can be described by the theoretical ‘random flight’-based equation, as follows:*k_a_*(*m*)/*k_a_(wt)* = (1 + Δ*N*/*N*)^−*n*^(1)
where *k_a_* (*m*) and *k_a_ (wt)* correspond to the respective deletion mutant and reference wild-type (ShB) association rate constants, while *N* and Δ*N* refer to ‘chain’ length and the ‘chain’ length difference, respectively [12], while *n* is a power law value. This equation was originally derived to explain fast channel inactivation described by a ‘ball and chain’ mechanism. In deriving this equation, the chain is assumed to be devoid of any structure and to function as an entropic ‘chain’ seeking the receptor site [12]. This equation could also fit the analogous ‘ball and chain mechanism for Kv channel binding to the PSD-95 scaffold protein. As can be seen in Figure 2c, a power law dependence with a value of 1.56 was determined as defining the relation between the normalized entry association rate constant and Kv channel C-terminal chain length [13]. Such an estimation of the *n* value (R^2^ = 0.96) is almost identical to the 3/2 theoretical power law value expected from polymer chain chemistry involving random flight of the ‘chain’ to allow the ‘ball’ to reach its receptor site [12]. The invariance between *k_a_* and *k_d_* of the Kv channel-PSD-95 interaction presented in Figure 2d further supports that the ‘timing’ criterion has been met in this case [13]. The same was found for the N-terminal inactivation channel ‘chain’ [17,18,23]. N-terminal ‘chain’ deletions were found to affect the kinetics of channel entry into the inactivated state [17,18], yet not the rate constant of channel recovery from the inactivation state [25].

Taken together, the studies reported here demonstrate how the three criteria that argue for a ‘ball and chain’ entropic clock-based mechanism have been fulfilled for both Kv channel fast inactivation and PSD-95-mediated clustering, implying that the mechanisms underlying these distinct processes are, in fact, analogous [13,34]. Nonetheless, the ‘ball and chain’ mechanism for scaffold protein binding is a molecular mechanism in essence and, therefore, one can speculate as to the physiological relevance of such an entropic clock mechanism with respect to the cellular clustering process. How such a mechanism might regulate channel properties dictating aspects of electrical signaling is described below.

## 3. The Physiological Relevance of Entropic Clocks in Kv Channel Inactivation and Clustering

The ‘ball and chain’ mechanisms of Kv channel fast inactivation and PSD-95-mediated channel clustering were further suggested based on studies involving deletion mutations in which the N- or C terminal chains were either shortened or extended [13,18]. These findings, however, raised questions about the physiological/cellular relevance of such an approach. For instance, once a channel is assembled, shortening of either of the N- and C-terminal chains is not possible. This obstacle can be overcome by considering the pattern of alternative splicing of the prototypical *Shaker* Kv channel gene. It turns out that alternative splicing of the *Shaker* gene occurs only at the edge segments encoding the N- and C-terminal ‘chains’ [35,36]. No splicing occurs at gene segments encoding the structured membrane-spanning voltage-sensing and pore domains, the electro-mechanical coupling of which leads to channel opening. Natural ‘chain’ variants differ only in their 5′ and 3′ exons, producing N- and/or C-terminal chains that vary in length and composition.

At the same time, spatial–temporal differences exist in the expression patterns of the N- or C-terminal spliced variant chains during development or in different tissues [35,36,37,38], further pointing to the physiological importance of alternative splicing in regulating channel inactivation or clustering, as governed by the ‘ball and chain’ mechanism. Indeed, natural N-terminal ‘chain’ splice variants were shown to exhibit distinct kinetics of channel entry into inactivation [39,40]. Changes in the kinetics of potassium current blockage through the Kv channel upon the ‘ball’ binding to its receptor site is expected to affect action potential shape, signal propagation, and firing frequency [4,5,6]. As for the other *Shaker* channel edge, the two short and long Kv channel C-terminal ‘chain’ splice variants (designated *A* and *B*, respectively) were shown to exhibit distinct affinities for PSD-95 and distinct clustering patterns [13]. The short ‘chain’ *A* variant presented higher affinity towards PSD-95, as compared to the long ‘chain’ *B* variant [13]. It further supported larger PSD-95-mediated channel cluster area sizes in the cell membrane, as presented in Figure 3. Thus, molecular distinctions reflected in the differential interactions of the tail variants with PSD-95 translate into functional differences in the context of cellular channel clustering [13]. Splicing-based changes in Kv channel cluster area size at unique membrane sites may subsequently lead to changes in action potential propagation and transmission [7,8,9,13,41].

Taken together, the *Shaker* Kv channel model protein system, with its functional N- and C-type ‘ball and chain’ inactivation and clustering mechanisms, exemplifies how linkage between alternative splicing and intrinsic disorder enables functional diversity to modulate ‘ball and chain’ interactions [42,43], in this particular case, in the context of neuronal electrical excitability.

In the following three sections, we provide further evidence for why we think that the ‘entropic clock’-based ‘ball and chain’ mechanisms describing N- and C-terminal-mediated inactivation and clustering processes are indeed analogous and emphasize the regulation aspect that such mechanisms bring to bear on electrical signaling. We begin by showing how a chimeric native-’chain’ approach has been used to argue that the ‘ball and chain’-based Kv channel clustering and inactivation mechanisms are parallel. We next consider how native channel C terminal ‘chain’ length was tuned to affect cluster Kv channel density. We end by demonstrating that this latter property can also be tuned by means of hereto-oligomerization subunit assembly. Regulation of Kv channel density in clustering sites is expected to affect action potential conduction properties.

## 4. A Native-‘Chain’ Chimeric Channel Strategy Supports Entropic Clock-Based ‘Ball and Chain’ Mechanisms for Kv Channel Clustering and Inactivation

The hallmark of ‘ball and chain’ mechanisms is the ‘chain’-length dependence of the appropriate binding process. Indeed, the ‘ball and chain’ mechanism for Kv channel inactivation and ‘PSD-95 binding’-mediated channel clustering were suggested based on artificial chain deletions [13,17,18]. To assess whether the N- and C-terminal cytoplasmic ‘chains’ of the Shaker Kv channel actually invoke a similar ‘ball and chain’ mechanism to regulate channel inactivation and clustering, the fact that alternative splicing of the Shaker Kv channel gene yields natural inactivation and clustering ‘chain’ length variants [35,36] which demonstrate length-dependent effects on channel inactivation [12,17,18] and channel PSD-95-binding [13], respectively, was exploited. These N- and C-terminal variant ‘chains’ can be used to construct chimeric channel variants with distinct N- and C-terminus swaps [34]. This native-’chain’ chimeric channel approach could validate the proposed analogy between the fast inactivation and clustering mechanisms. In a nutshell, should ‘chain’-level chimeric Kv channels in which the wild-type N-terminal inactivation or C-terminal clustering chain sequences were exchanged with natural ‘chain’ sequences normally found at the opposing end of the molecule (Figure 4a) still support channel inactivation or clustering (as appropriate) in a length-dependent manner, this would be a clear indication that unrelated Kv channel inactivation and clustering also rely on a ‘ball and chain’ binding mechanism involving extended random chains.

The analysis confirmed this scenario. We found that native spliced variant fast inactivation and clustering ‘chains’ could both replace their counterpart when attached to the appropriate ‘ball’ sequence motif [34]. Specifically, when the native inactivation chains replaced the original clustering ‘chain’ variants and were fused to the clustering ‘ball’, the affinity of the chains to PSD-95 presented the expected linear dependence on ‘chain’ length (not shown). Furthermore, the inactivation ‘chains’ showed power law dependence of the rate constant for channel-PSD-95 binding on chain length (Figure 4b), suggestive of the inactivation ‘chains’, like their clustering counterparts, also timing complex formation, as predicted by a random flight description [12]. In accordance, the effect of ‘chain’ length on the binding was observed for the association rate constant alone (Figure 4c) [34]. When considering the full-length channel, the inactivation ‘chains’ also supported PSD-95-mediated *Shaker* channel clustering in a length-dependent manner (not shown; see reference [34]). We next examined fast N-type channel inactivation of WT and chimeric channels in which the original N terminal inactivation ‘chains’ were replaced by native clustering ‘chains’ still fused to the inactivation ‘ball’. We found that WT and chimeric channels support fast N-type channel inactivation in a length-dependent manner [34]. Here, too, the scaled entry into an inactivation rate constant (*k*on) of the clustering ‘chains’ adhered to the power law dependence on chain length difference with a value close to 3/2 for all inactivation and clustering chains considered (Figure 4d). Again, invariance between *k*on and *k*off was observed (Figure 4e). This correlation plot thus adheres to the kinetics ‘timing’ criterion of entropic clock-based binding mechanisms [10,13,34].

Taken together, the chimeric channel strategy using native clustering and inactivation chains [34] yielded results in agreement with results reported using artificially deleted ‘chains’ [13,17,18], with both sets of measurements demonstrating ‘chain’ length-dependent binding kinetics. Both approaches involve entropic ‘chains’ that do not fold upon binding and provide only the degrees of freedom required to reach the ‘ball’ receptor site [12,13]. Moreover, inactivation and PSD-95-binding, assessed using either native or artificial ‘chains’, revealed invariance between the corresponding forward and backwards rate constants (Figure 4c,e), thereby presenting expected timing signatures of ‘entropic clock’-based binding [34]. The two ‘ball and chain’ mechanisms for fast inactivation and PSD-95 binding can thus be considered analogous. One could, however, argue that differences in the molecularity of the binding reactions, with that of fast inactivation being 1st order and that of the clustering processes being 2nd order, undermine such an assertion. Still, the parallel between the N- and C-terminal-based ‘ball and chain’ mechanisms is further supported by considering that PSD-95 is a priori membrane-associated as a result of its palmitoylation [44], as well as the finding that the onset of Kv channel fast inactivation transpires similarly when the ‘ball and chain’ sequence is found on the auxiliary β subunit of the channel with which it interacts, rather than on the channel per se [45]. This variation on fast inactivation argues that both scaffold protein-binding and channel inactivation are timed by entropic chains that rely on a similar ‘ball and chain’ mechanism [34].

## 5. Cellular Clustering Correlates of the PSD-95-Kv Channel ‘Ball and Chain’ Binding Mechanism

The ‘ball and chain’ mechanism describing Kv channel-PSD-95 interaction is molecular in essence, and thus provides no information on Kv channel clustering. This raises the question of the identity of cellular correlates of the ‘ball and chain’ mechanism for channel–scaffold protein binding, if they indeed exist (Figure 1c). Earlier work from our lab reported that Kv channel ’chain’ length determines cluster area size [13,34] (Figure 3a). Still, it remains unknown whether or not Kv channel ‘chain’ length regulates Kv channel density (Figure 5a). With this question in mind, we recently conducted sub-diffraction high-resolution confocal imaging microscopy of PSD-95-mediated Kv channel clustering, along with quantitative clustering analysis, to determine cluster ion channel densities [46]. We found that the short and long *A* and *B* spliced variant ‘chains’ exhibited distinct clustering phenotypes. In particular, the two ‘chain’-length channel variants exhibited differences in the mean number of clusters per cell and the fraction of channels in clusters, with the short ‘chain’-bearing high-affinity *A* variant presenting higher values than the long chain-presenting low-affinity *B* variant. Furthermore, the two variants exhibited distinct clustering density histograms, with that of the shorter, high-affinity (to PSD-95) *A* variant being skewed to higher channel density regimes (Figure 5b,c). Moreover, our study revealed that systematic shortening of the Kv channel C-terminal ‘chain’ determined cluster Kv channel density with bell-shaped dependence, reflecting a balance between steric hindrance and thermodynamic considerations controlling ‘chain’ recruitment by the PSD-95 scaffold protein (Figure 5d). Systematically shortening the Kv channel ‘chain’ monotonically increased the magnitude Kv channel density within clusters up to a certain ‘chain’ length, from which point further shortening resulted in the opposite trend. Considering the stoichiometry of the channel–scaffold protein interaction, involving the four ion channel tails and three different PDZ domains of PSD-95, together with plausible models for channel clustering [14], one can conclude that such behavior reflects steric hindrance resulting from the inability to crown several channel molecules in close proximity on the same of PSD-95 molecule when ‘chain’ length is too short [46]. The bell-shaped impact on channel density on ‘chain’ length recalls the effect of N-terminal ‘chain’ shortening on Kv channel fast inactivation. There, shortening the ‘chain’ increased the rate of channel entry into the fast inactivated state, with too-short ‘chains’ reducing the rate of inactivation simply because the inactivation ‘ball’ motif found it harder to reach its receptor site in the open channel pore [12,18].

Together, these findings demonstrate the cellular correlates of the molecular ‘ball and chain’ mechanism for channel-PSD-95 binding, with respect to channel clustering. Ion channels density within clusters, as well as other clustering properties are affected by *Shaker* Kv channel C-terminal ‘chain’ length [46]. This further strengthens the argument for the existence of entropy-based regulation of Kv channel clustering that mirrors the thermodynamic entropy profile of the earlier Kv channel-PSD-95 interaction [13,34]. Evidence for a timing aspect of cluster formation is still not available.

## 6. Subunit Hereto-Oligomerization Regulates Cluster Kv Channel Density

Homo-tetrameric *Shaker* Kv channels, composed of only one splicing variant, either the low-affinity *B* or the high-affinity *A* subunit, exhibit distinct channel clustering densities (Figure 5b,c) [46]. Such regulation of channel clustering is carried out through distinct interactions of the short (*A*) or long (*B*) C-terminal channel ‘chain’ with PSD-95. Considering that both *Shaker* clustering variants are expressed during similar developmental stages and in similar fly tissues [37,38] and that both variants exhibit an identical T1 assembly domain [35,36], one can reasonably assume that heterotetrameric channels bearing different numbers of the high- (*A*) and low-affinity (*B*) subunits could assemble, possibly leading to the appearance of distinct densities of Kv channels within clusters (Figure 6a). Such regulation might be expected, given that heterologous assembly of different N-terminal ‘chain’ spliced variant subunits was found to regulate N-type fast inactivation [47]. Such a means of regulation is not expected to be prone to steric hindrance (as revealed by systematic deletion of the C-terminal ‘chain’ [46]), as the native variants generated are not so short. In the case of such regulation, each different combination of *Shaker A* and *B* subunits in a heterotetrameric channel can give rise to distinct affinity for PSD-95, subsequently resulting in distinct channel clustering density (see Figure 6a).

Indeed, the native *A* and *B* ‘chain’ subunit variants of the *Shaker* Kv channel assembling into heterotetrameric channels present different numbers of both variant subunits [48]. Such inter-subunit assembly was confirmed in both the molecular and cellular contexts. To address the possible effect of hetero-oligomeric subunit assembly on channel clustering densities, different ratios of the variants were tested by manipulating the ratio of the high-affinity *A* subunits within the total amount of *Shaker* channel DNA transfected to cells (*f*A), and cell imaging was subsequently performed. Interestingly, employing different *A:B* DNA transfection ratios (i.e., higher *f*A values) led to similar channel cell surface expression. Hence, the expression of both variants appears to be unbiased. In other words, changing channel affinity toward PSD-95 as a result of changes in ‘chain’ length or in the number of high-affinity *A* subunits within the heteromeric channel, fails to affect channel expression levels [48].

However, clustering site-level statistical analysis showed that, while the sizes of the cluster area did not differ as a function of the transfection ratios employed, changes in the mean value of Kv channel density within clusters were seen (Figure 6b) [48]. A linear empirical correlation between the two quantities was recognized, with higher *f*A values (i.e., higher *A:B* DNA transfection ratios) yielding progressively greater cluster Kv channel densities (Figure 6b). Thus, augmenting the weighted affinity of all heterologous Kv channel subunit combinations presenting higher numbers of the *A* subunit yielded increased densities of Kv channel molecules within clustering sites, reflecting higher channel molecule copy numbers. This finding was further supported by the monotonic linear dependence of cluster Kv channel density of the homo-tetrameric *B* channel and the affinity of the channel-PSD interaction pair, as affected upon systematic shortening of the *B* channel C-terminal region [46]. Here, however, artificial over-shortening of the ‘chain’ resulted in a lowering of cluster Kv channel density as the result of steric hindrance, related to the inability of the multiple PDZ domains of PSD-95 to bind adjacent channel molecules bearing tails that were too short. In nature, such very short chains are not found, with no bell-shaped behavior being observed upon heterologous channel assembly [48].

To summarize, heterologous subunit assembly provides a vehicle for regulating cluster Kv channel density. Specifically, spatial and temporal tuning short (*A*) or long (*B*) or of both C-terminal native ‘chain’ variant levels [37,38] would permit the introduction of channel variability in hetero-oligomeric subunit assembly [47,48] and generate channels with distinct affinities to PSD-95, subsequently leading to distinct PSD-95-mediated Kv channel densities. Such modulation would cause changes in current density at sites of homo- or hetero-oligomeric channel localization, for example, at the post-synaptic density, as demonstrated here. This could result in changes in action potential transmission and frequency, as well as in synaptic growth and plasticity [7,8,9].

## 7. Concluding Remarks

This article addressed the entropic clock function of intrinsically disordered protein segments, as reflected by the ‘ball and chain’ mechanisms associated with ion channel activity. Such ion channel regulation mechanisms have a profound impact on electrical signaling. An entropic chain-based ‘ball and chain’ mechanism was initially proposed in the 1970s for voltage-dependent sodium channel inactivation [16] and subsequently for Kv channel fast inactivation [17,18], when disordered protein segments were only beginning to be recognized. The idea of intrinsically disordered protein segments, with their diverse functional roles, was not yet developed, nor were analysis methods to assess the unstructured state of such segments at the time. The various roles played by intrinsically disordered protein segments in almost all cellular processes [11,49,50,51,52,53,54,55] support the no-structure–function scenario that complements the traditional structure–function narrative [50,53,54,55].

The two distinct entropic clock-based binding mechanisms discussed here for Kv channel inactivation and clustering share a unique thermodynamic signature [10,34] and are modulated by alternative *Shaker* Kv channel gene splicing to yield N- or C-terminal chain length variants [35,36], giving rise to hetero-oligomeric channels presenting distinct inactivation kinetics or ion channel densities [48], respectively. Such modulation, by mean of entropic clock protein chains, has been implicated in the operation of different mammalian ion channels. A mutation in the human Kv 1.1 channel gene that resulted in deletion of the entire channel C-terminal tail was assigned responsibility for the onset of episodic ataxia type 1 [56]. Electrophysiological whole-cell recordings served to demonstrate how this mutation led to a dramatic drop in potassium current, an observation in accordance with the likely loss of interactions with scaffold proteins, thought to compromise channel clustering. Thus, modulating the density of Kv channel within clustering sites by means of alternative splicing modulates channel electrical properties by affecting underlying action potential shape, propagation, and transmission. Malfunctions associated with such entropic clock ‘chains’ can thus cause nervous system dysfunction [56].

## Figures and Tables

**Figure 1 entropy-25-01351-f001:**
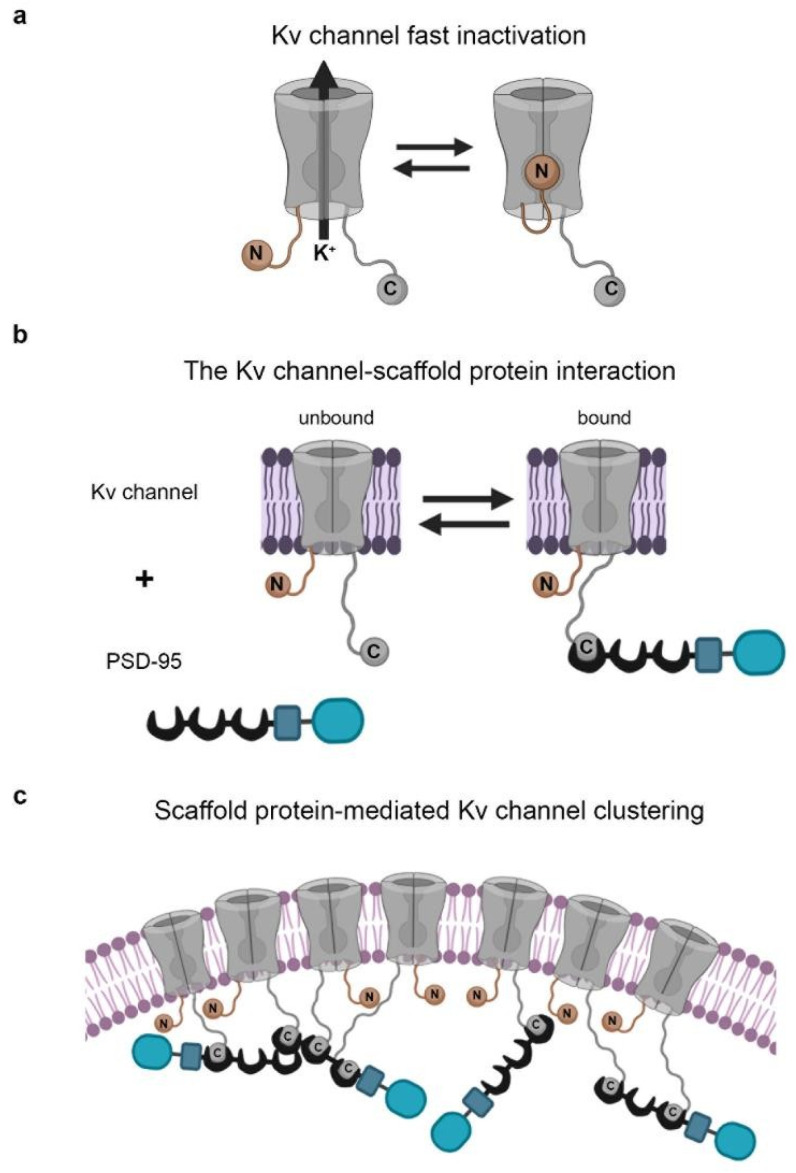
Entropic clock-based ‘ball and chain’ mechanisms of Kv channel inactivation gating and clustering. (**a**) Schematic representations of the ‘ball and chain’ mechanisms for channel fast inactivation. In fast inactivation, the open (O) Kv channel pore inactivates (I) in a precisely timed manner, as determined by N-terminal chain length, upon binding of the ‘chain’-tethered ‘ball’ to a receptor site within the inner cavity of the open pore. (**b**) Schematic representation of the ‘ball and chain’-like mechanism for Kv channel binding to the PSD-95 scaffold protein. In this mechanism, the extended ‘chain’ at the channel C-terminal binds PSD-95 upon interaction of the chain-tethered peptide ‘ball’ with PSD-95 PDZ domain(s) in a manner reminiscent of the process of fast (N-type) channel inactivation. The Kv channel-PSD-95 interaction is thus entropy-controlled and dictated by Kv channel ‘chain’ length. Given the stoichiometry of the interaction [14] and the ability of PSD-95 to aggregate [15], channel clustering occurs (**c**). The membrane-embedded portion of the Kv channel corresponds to the voltage-sensor and pore domains. The crescent, rectangular and rounded box shapes represent the PDZ, SH3 and guanylate kinase-like domains of the PSD-95 protein, respectively. This and all other figures were prepared using the BioRender scientific image and illustration software.

**Figure 2 entropy-25-01351-f002:**
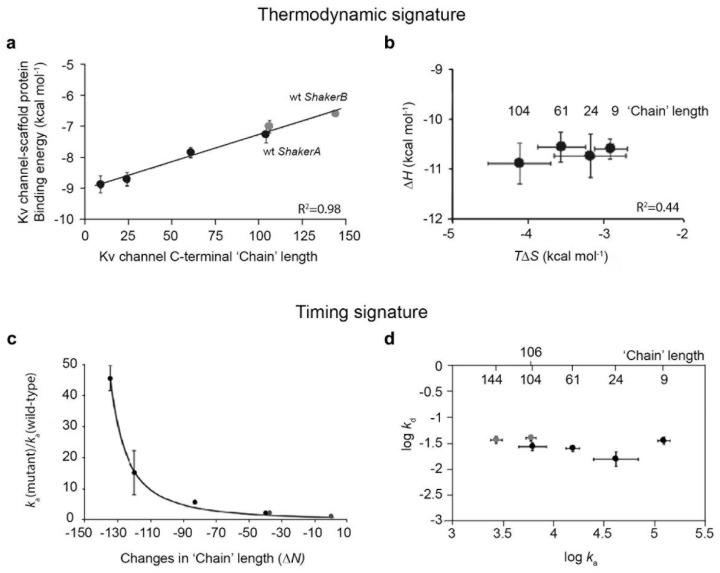
Thermodynamic and kinetics signatures of entropic clock-based ‘ball and chain’ binding mechanisms. (**a**) Plot of the association binding energy between natural and artificially shortened *Shaker* channel C-terminal ‘chains’ and the PSD-95 scaffold protein as a function of ‘chain’ length. The wild-type *Shaker A* and *Shaker B* ‘chains’ are indicated by grey symbols. (**b**) Enthalpy–entropy correlation plot of the *Shaker* B-PSD-95 scaffold protein interaction. (**c**) Dependence of the scaled *Shaker* A/B-PSD-95 association rate constant (*k_a_* mutant/*k_a_* wild type. Δ*Shaker* B) on chain length difference (Δ*N*). The solid line represents the best fit of the data to Equation (1) (see text) describing a random flight ‘ball and chain’ mechanism for channel ‘chain’ binding to PSD-95. The values for the wild-type *Shaker A* and *Shaker B* C-terminal ‘chains’ are indicated by grey symbols. (**d**) Association–dissociation rate constant correlation plot of the *Shaker A/B*–PSD-95 interaction. The wild-type *Shaker A* and *Shaker B* C terminal ‘chains’ are indicated by grey symbols. Numbers on the upper horizontal axis indicate ‘chain’ length. Data presented in this figure were compiled from reference [10]. (see details therein).

**Figure 3 entropy-25-01351-f003:**
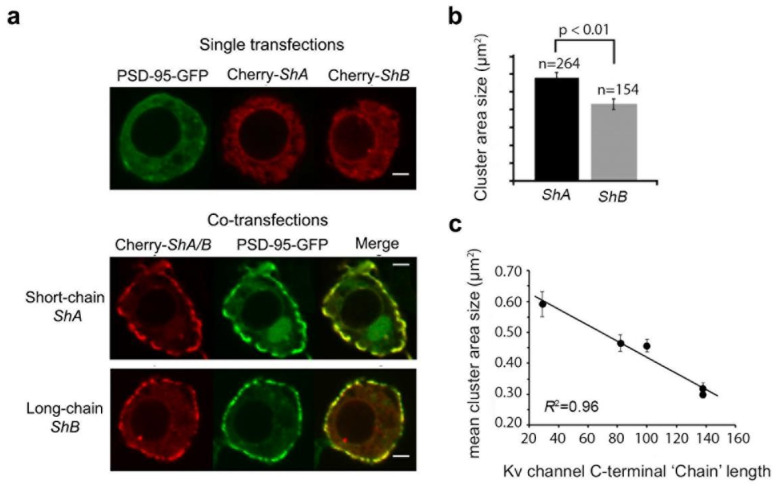
Distinct clustering phenotypes of the native *A* and *B Shaker* Kv channel ‘chain’ length variants. (**a**) High-resolution confocal microscopy images of *Drosophila* S2 Schneider cells expressing PSD-95–GFP or either of the *Shaker* A or *B* ‘chain’ length variants alone (upper panel). In the lower panel, high-resolution confocal microscopy images of Schneider cells co-expressing PSD-95–GFP and either the mCherry-*Shaker A (ShA)* or mCherry-*Shaker B (ShB)* are presented. Three images are shown for each cell, with the red channel-associated fluorescent signal shown in the left column, the green PSD-95-associated signal presented in the middle column, and the merged shown in the right column. Scale bars correspond to 2 mm. (**b**) Comparison of averaged cluster area size of *A* and *B* channel clusters (*n* = 150–260; *p* < 0.01 in Wald χ^2^ test). (**c**) Dependence of the mean cluster area size of the different channel variants on C-terminal ‘chain’ length. Error bars in the appropriate figure panels represent SEM values [34].

**Figure 4 entropy-25-01351-f004:**
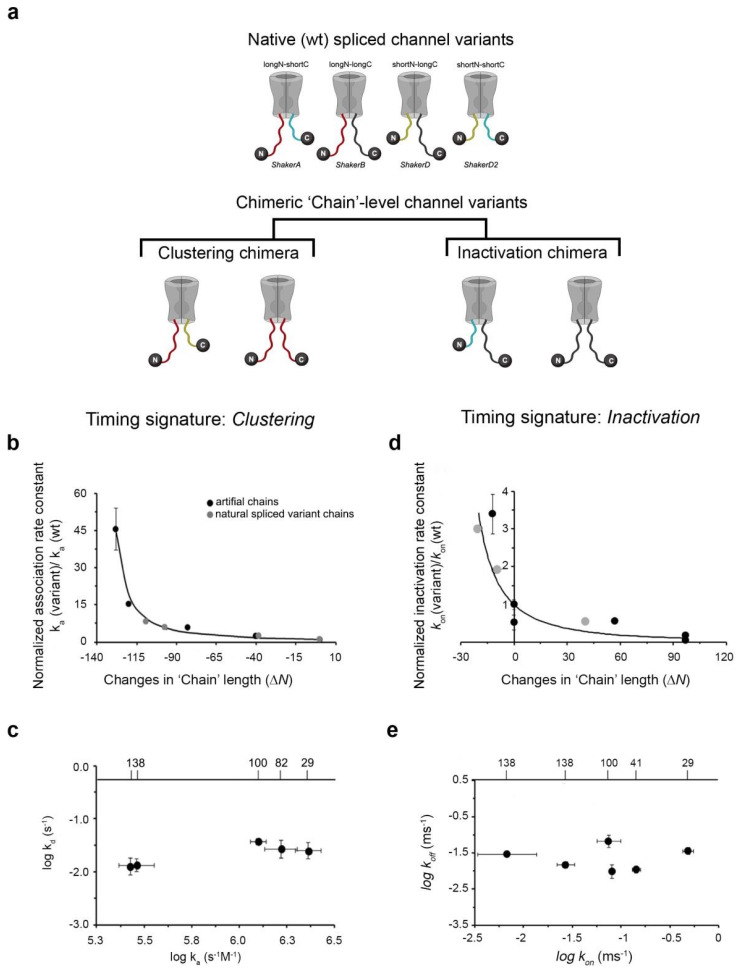
Direct evidence for a similar ‘ball and chain’ mechanism underlying *Shaker* Kv channel inactivation and clustering. (**a**) Schematic representation of wild-type *Shaker* channel variants and of the chimeric native channel ‘chain’-level approach used to assess the compatibility of the inactivation and clustering ‘ball and chain’ binding mechanisms [34]. The four wild type Kv channel spliced variants used in the current study share an identical core T1 (rectangular shape) and membrane-spanning regions (voltage-sensing and pore domains) but different N- and C-terminal ends. The variants represent the four possible combinations of channels containing either long (L) or short (S) inactivation (I) chains at the N-terminus (respectively denoted by red and yellow) and long and short clustering (C) ‘chains’ at the C-terminal tail (denoted by black and blue). Chimeric channels were generated by replacing either the inactivation or clustering ‘chain’ with the short or long native ‘chain’ from the other end of the channel. In all chimeric channels, the original ‘ball’ motif remained intact, thus ensuring that inactivation and clustering occurred at the appropriate N- or C-terminal end. (**b**) Native short and long inactivation and clustering ‘chains’ (all attached to the same clustering ‘ball’) were examined using SPR analysis to assess binding to the second PDZ domain of PSD-95. The normalized association rate constant for each case (*ka* /*ka* reference (*ShB*)) was plotted against chain length difference (∆*N*). Gray data points represent artificial ‘chain’ deletions, whereas black points refer to native clustering for inactivation ‘chain’ swaps. The solid line represents the best fit of the data to Equation (1) with the expected power law dependence (see text). (**c**) Association–dissociation rate constant correlation plot of the native ‘chain’-PDZ2 interaction. Numbers on the upper horizontal axis indicate native ‘chain’ length. (**d**) Kv channels with different N terminal native ‘chains’ (either long or short clustering or inactivation ‘chains’ and all with similar inactivation ‘ball’) were examined using a recording protocol to evaluate fast N-type channel inactivation. The panel describes the dependence of the scaled forward inactivation rate constant (*kon*/*kon* reference (*ShA*)) of native wild type channels (*A* or *B*), chimeric channel inactivation to clustering ‘chain’ swaps (both in black symbols) and ‘Chain’ deletion or insertion mutants (gray symbols) on the ‘chain’ length difference (∆*N*). The solid line represents the best fit of the data to Equation (1) with the expected power law dependence of 3/2 (see text). (**e**) Correlation plot relating the backward (*k*off) and forward (*k*on) inactivation rate constants of the different wild-type or inactivation chimera channel proteins. Numbers on the upper horizontal axis indicate ‘chain’ length. See reference [34] for further details on all figure panels.

**Figure 5 entropy-25-01351-f005:**
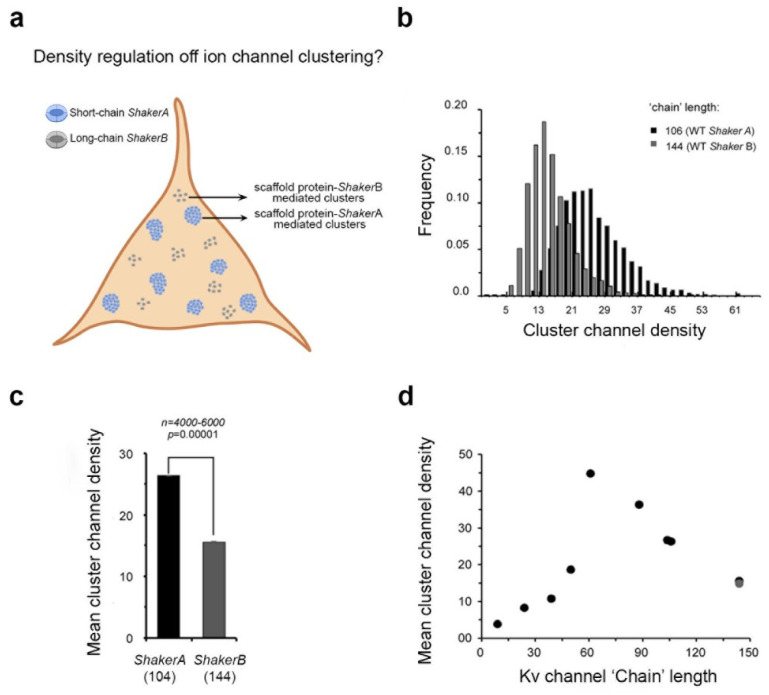
Molecular and cellular correlates in Kv channel clustering: Entropy-based regulation of cluster ion channel density. (**a**) Schematic representation of possible changes in cluster Kv channel densities due to changes in ‘chain’ length. (**b**) Cluster Kv channel density distributions of the Shaker *A* and *B* native channel variants (evaluated by cluster signal intensity), as supported by PSD-95. (**c**) Comparison of the mean value of PSD-95-mediated *A* or *B* cluster Kv channel density. (**d**) Dependence of the mean value of PSD-95-mediated cluster Kv channel density on C-terminal ‘chain’ length. Differences in cluster ion channel density were all found to be statistically significant, based on an ANOVA test (*n* = 350–7500; *p* < 0.00002). See reference [46] for further details.

**Figure 6 entropy-25-01351-f006:**
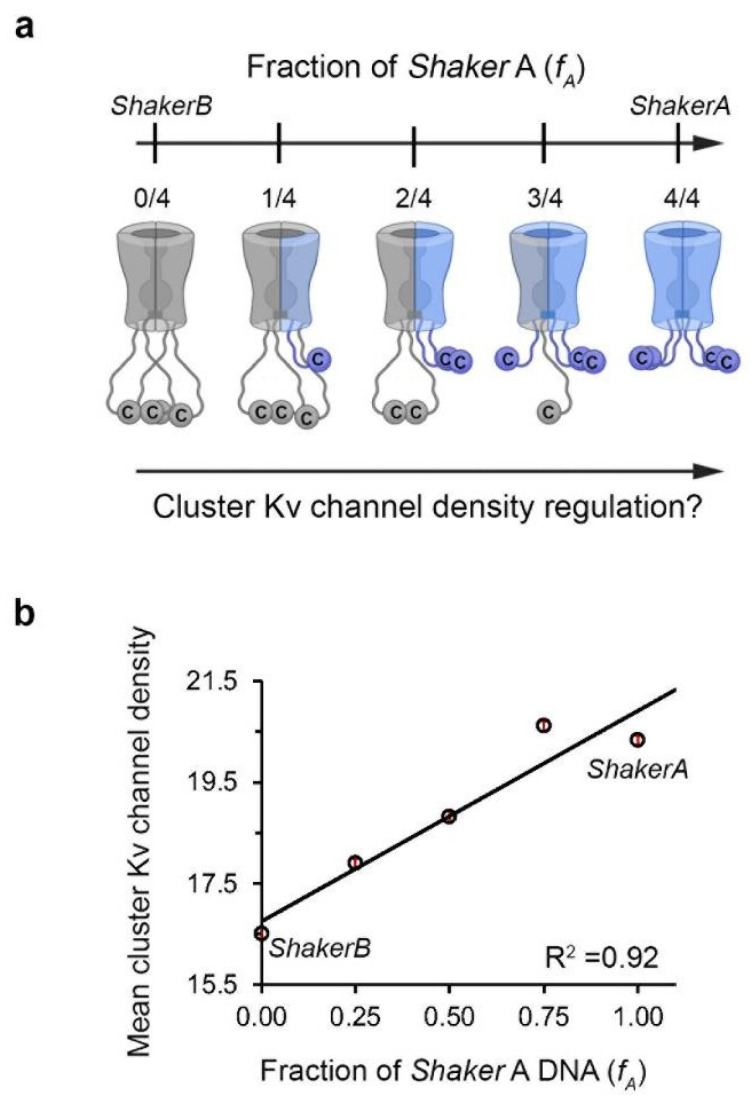
Regulating *Shaker* Kv channel clustering by hetero-oligomerization subunit assembly. (**a**) Heterologous subunit assembly of both *Shaker A* (light blue) and *B* (gray) variants can produce tetrameric channels with different numbers of the high-affinity *A* (or *B*) subunits, potentially giving rise to distinct PSD-95-Kv channel affinities and cluster Kv channel densities. For clarity, the intrinsically disordered N-terminal inactivation chains were omitted from the channel drawings. (**b**) Dependence of the mean value of PSD-95-mediated cluster Kv channel density on the *A:B* subunit DNA transfection ratio. Differences in cluster ion channel density were all found to be statistically significant, based on ANOVA (*n* = ~3000–5000; *p* < 0.0001). See reference [48] for further details.

## Data Availability

Not applicable.

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
