# Peer review of "It’s Time for Entropic Clocks: The Roles of Random Chain Protein Sequences in Timing Ion Channel Processes Underlying Action Potential Properties"

_entropy, 2023, doi:10.3390/e25091351_

Round 1
Reviewer 1 Report
The review paper summarizes current advances in the understanding of the role of the N- and C-terminal, intrinsically disordered protein segments in voltage-gated K+ channels. Several original papers, mostly for the laboratory of the senior author, have indicated that N-type inactivation and ion channel clustering are controlled by the entropic clocks established by the aforementioned disordered protein segments.
The review is very clearly structured and gives the reader a comprehensive summary of the field. The paper uses several illustrations to facilitate the understanding of the topic.
The paper is well written, however, I do have some comments for the authors that they might consider for the improvement of the review:
The authors are correct that N-type inactivation for voltage-gated K+ channels was described by Aldrich and colleagues however, the original concept of N-type inactivation came from Clay Armstrong and Pancho Bezanilla. I would like to ask the authors to incorporate this information into the manuscript.
Page 3, 1st paragraph: ..and can resulted in…”, please change it to “…can result in…”
Page 3, 3rd paragraph: “Inactivation refers to the process… “: inactivation is not specific to K+ channels, please rephrase
Page 3, 3rd paragraph, end of page: “caused accelerated channel entry into the inactivated state”. please change it to “caused accelerated entry of the channels into the inactivated state”
Page 4, last paragraph: Please refer to Fig. 2 A-B in the last paragraph, without the reference it is difficult the imagine the delta H and delta S components. Moreover, I think that the discussion of the contributions to delta H and Delta S should be completely moved to the next page, where the details of Fig. 2 are. In the current version the discussion is started, and then detailed in the next chapter. I think that for the second point this text is enough: “Second, the binding of the 'ball' sequence to its receptor site should be entropy- controlled, with the ‘chain’ length determining the binding energy of the reaction (see below, Fig. 2).”
Fig. 2: please indicate where data are from. The current figure legend is not clearly stating that all data are from ref 17, someone might understand that only panel D data are.
Page 4 equation: define “n” please (later there is reference to the exponential term but should be defined where the equation is)
Page 4 2nd paragraph: I think that citation of ref 28 is wrong. It is about the clustering, not N-type inactivation
Page 6 (and later) The authors use the term “the invariance between…”. I am not sure that invariance is the best term to represent that ka depends on some variable whereas kd does not. Please check if this is the right English expression for this.
Fig: 4: please provide a full and detailed description of Fig 4. The current shortened version may only be understood if one knows the original paper. Color coding is difficult to see, and is not explained in the figure legends. Please use colors that are easier to distinguish, or use thicker lines or both. (gray, gold and blue are almost the same to me on the figure, I needed a lot of time to understand what the different colors meant).
Page 9: rewrite the first sentence please, quite confusing, (replaced fused??): “As for the testing the opposite configuration of the native clustering 'chains', whereby the original N-terminal inactivation chains were replaced fused to the inactivation 'ball',”
Page 9: “The parallel between the N- and C-terminal-based ‘ball and chain’ mechanisms is further supported upon….” How does the rest of the sentence support the parallels?
Page 10: “We found that the short and long A and B spliced variant 'chains' exhibited distinct clustering phenotypes (not shown).” Even if it is not shown please explain what you mean by distinct. This way the sentence does not give any information to the reader.
Page 11 (and in several other places in the manuscript): Please check in the literature the writing of “hetero-tetrameric”. I have seen it mostly written as “heterotetrameric”.
Page 11: This is difficult to understand: “…leading to the appearance of distinct cluster Kv channel densities (Fig. 6a), much like the assembly of different inactivation splice variant-generated subunits [48].” What does “distinct cluster Kv channel densities” mean? Density of Kv channels in the cluster? Difference in the densities of the clusters? Please clarify this. This way it is unclear. The same applies to the second part of the sentence: “inactivation splice variant-generated subunits”? Please write more clearly this.
Page 11: “Indeed, the native A and B 'chain' subunit variants of the Shaker Kv channel assembles into hetero- tetrameric channels presenting different numbers of both variant subunits [49].” I assume that the meaning of this sentence is the following: Indeed, the native A and B 'chain' subunit variants of the Shaker Kv channel assembling into heterotetrameric channels present different numbers of both variant subunits [49]. If this was the intention then use the sentence suggested here.
Page 11: “different ratios of the variants were tested by regulating the ratio of the high-affinity …” I would recommend to use the word “manipulating” then “regulating”. What was done is mixing various ratios of DNA i.e. playing with the proportions, rather than regulation for example the expression of subunits: “different ratios of the variants were tested by manipulating the ratio of the high-affinity A”.
Page 11: At first this is written: “ratio of the high-affinity A subunits within the total amount of Shaker channel DNA “ which translates to A/(A+B) to me. Later A:B ratio is used in the text but A/(A+B) is used in the figure (i.e., 4/4 rather than 4:0). Please harmonize the nomenclature throughout the manuscript.
Page 11: this sentence is difficult to read and comprehend: “However, clustering site-level statistical analysis showed that while the mean cluster area size did not differ as a function of the transfection ratios employed, changes in the mean value of cluster Kv channel density were seen (Fig. 6b) [49]”. Based on what I understood the following sentence would be better:
“However, clustering site-level statistical analysis showed that while size of cluster area did not differ as a function of the transfection ratios employed, changes in the mean value of Kv channel density within the clusters were seen (Fig. 6b) [49]. Please revise accordingly.
Page 12: Too long and complicated sentence: “Thus, augmenting the weighted affinity of all heterologous Kv channel subunit combinations presenting higher numbers of the A subunit, realized upon raising the A:B transfection ratio, yielded increased densities of Kv channel molecules within clustering sites, reflective of higher copy numbers of channel molecules.” Almost 4 lines! Please rephrase.
Page 12: …such ion channel mechanisms…” I guess it should be “such ion channel regulatory mechanisms”
Page 13: “in the workings of ion channels” I would recommend “in the operation of ion channels”
Page 13: “cluster Kv channel density” please see my comments above, I think that this term is impossible to understand. Is it the density of Kv in the clusters?
Simplify long sentences please and describe the terms used in the manuscript more clearly as indicated in the review
Reviewer 2 Report
The authors should be congratulated for this excellent review discussing molecular mechanisms of entropic clocks. The manuscript is well-written, concise, and presented in a very logical way. It contains a lot of useful information and will be of interest to the readers of the journal. I really enjoyed reading it and think that the authors did an outstanding job here.
Author Response
Thank You!